# Healthcare providers' palliative care graded diagnosis and treatment behavior, attitudes, self-efficacy, compassion fatigue, and workplace well-being: A mediating moderation model

Lixiang Liu[1☉‡], Meidi Xiong[2,3☉‡], Fang Hong[4], Yiting Zhang[5], Chunhua Zhang[iD][3]*

**1** Department of Head, Neck and Pediatric Oncology, Hubei Key Laboratory of Tumor Biological Behavior, Second Clinical College of Wuhan University, Hubei Province Cancer Clinical Study Center, Zhongnan Hospital of Wuhan University, Wuhan, China, **2** Zhongnan Hospital of Wuhan University, Wuhan, China, **3** Nursing Department, Zhongnan Hospital of Wuhan University, Wuhan, China, **4** Yangxin County People's Hospital, HuangShi, China, **5** Huangshi Maternity and Children's Health Hospital, HuangShi, China

☉ These authors contributed equally.
‡ Lixiang Liu is First author, Meidi Xiong is Co-first author.
* zhangchua@whu.edu.cn

## Abstract

### Background

Palliative care is essential for end-of-life (EOL) patient care. While prior research has acknowledged the role of self-efficacy in nursing, its specific mechanisms within graded diagnosis and treatment contexts remain underexplored.

### Aims

(1) To investigate the relationship between healthcare providers' attitudes and behaviors regarding palliative care, graded diagnosis, treatment and the mediating role of self-efficacy. (2) To explore whether self-efficacy moderates the effect of compassion fatigue on well-being at work.

### Design

A cross-sectional study was conducted at two tertiary hospitals in Hubei Province, China, and data were collected from 900 healthcare providers in July and August 2023.

### Methods

Four validated self-report scales (Palliative Care Graded Diagnosis and Treatment Knowledge, Attitude, and Behavior scale, Self-Efficacy Scale, Brief Compassion Fatigue Scale, and Nurses' Well-being at Work Scale) were used to collect data. Analyses were performed using SPSS PROCESS.

**Data availability statement:** The dataset from which the results of the study were produced (Supplementary Information Data). All relevant data are within the paper and its Supporting Information files.

**Funding:** This work was Supported by Special Research Cultivation Fund for Clinical Nursing of Wuhan University, Project LCHL202327. The funders had no role in the study design, data collection and analysis, decision to publish, or manuscript preparation.

**Competing interests:** The authors report no declarations of interest.

**Abbreviations:** EOL, End-of-life; KAP, Knowledge, Attitude, and Practice; ASCO, American Society of Clinical Oncology; SEM, Structural Equation Modeling; GSES, The General Self-Efficacy Scale

## Results

Significant correlations were found among healthcare providers' attitudes toward graded diagnosis and treatment, self-efficacy, and their behaviors. Healthcare providers' attitudes towards graded diagnosis and treatment predicted an increase in self-efficacy ($\beta = 0.161$, $p < 0.001$), which subsequently led to improved graded diagnosis and treatment behavior ($\beta = 0.647$, $p < 0.001$). Self-efficacy mediated 34.81% of the effect of attitudes on graded diagnosis and treatment behavior. Although significant correlations existed among healthcare providers' compassion fatigue, self-efficacy, and well-being at work, further analysis revealed that self-efficacy played a moderating role.

## Conclusion

Self-efficacy plays a crucial role in palliative care graded diagnosis and treatment, moderating the relationship between compassion fatigue and well-being at work. This finding indicates that enhancing self-efficacy not only improves nursing practice in palliative care but also alleviates emotional stress and boosts professional well-being.

## Background

In the European Union, the proportion of the population aged 65 and over has already been surpassed by the share of those aged under 15, and it is anticipated that by 2060, this share will have more than doubled. The increase in the number of older individuals is forecasted to be even more pronounced among those at advanced ages [1]. China has entered a stage of moderate aging, with a population of 190.64 million aged 65 and above, while the incidence of chronic diseases and mortality from malignant tumors remains high [2,3]. This further exacerbates the need for palliative care for patients in both healthcare and community settings. The American Society of Clinical Oncology (ASCO) guidelines now position palliative care alongside surgery, radiation therapy, and chemotherapy as integral components of cancer management [4]. Palliative care has played an indispensable role in patients' end-of-life(EOL) care.

However, the issue of global healthcare inequality is particularly pronounced, with lower-and middle-income countries facing significantly greater challenges compared to high-income nations [5]. For instance, the average palliative care coverage is 41%, but two regions of Colombia have a coverage below 30% [6]. In China, fewer than 1% of terminally ill patients have access to palliative care services [7]. The quality of EOL care for patients has become a pressing issue in China. Palliative care encompasses EOL support provided by healthcare providers and social workers, offering medical and psychological assistance. It aims to relieve pain and other distressing symptoms in terminally ill patients, thereby fostering peace, comfort, and dignity, while offering supportive services to families [8]. In the terminal stages of life, patients often endure pain, fatigue, depression, and anxiety, which challenge their ability to

maintain autonomy, dignity, and a sense of control. Over-medicalization may worsen suffering instead of improving the quality of life [9]. Palliative care prioritizes improving the quality of life and preserving dignity for patients, rather than solely extending life. Its focus on assisting patients in a comfortable, peaceful, and dignified EOL journey, while also offering support services for their families.

In China, a graded diagnosis and treatment system is implemented, whereby medical cases are categorized based on disease severity and treatment complexity. Research has demonstrated the pivotal role of graded diagnosis and treatment behavior in the implementation of palliative care [10]. A tripartite palliative care center has been established to integrate hospitals, communities, and home care within the graded diagnosis and treatment system [11]. This approach helps maximize regional resources by closely combining major hospitals with grassroots healthcare facilities, fostering vertical and horizontal coordination. By harnessing the medical resources, technology, and manpower advantages of large comprehensive hospitals, and combining them with the convenient location and ample bed capacity of grassroots hospitals [10]. This approach can alleviate overcrowding in large hospitals and underutilization of beds in community healthcare institutions while providing personalized medical services. Through graded diagnosis and treatment behavior, a more accurate assessment of patients' conditions can be made, leading to the development of targeted treatment plans and improvements in treatment effectiveness and quality of life. Therefore, identifying factors that influence graded diagnosis and treatment behavior and exploring their potential mechanisms in palliative care are crucial.

Of particular significance is the pivotal role of healthcare providers as the primary providers of palliative care graded diagnosis and treatment, with their attitudes directly shaping the efficacy of such treatment. Both cross-sectional and longitudinal studies have underscored the protective influence of healthcare providers' positive attitudes towards palliative care graded diagnosis and treatment on their conduct during the process. Research indicates that the willingness to serve, level of knowledge, and communication skills among healthcare providers have a direct bearing on the effectiveness of palliative care graded diagnosis and treatment [12]. However, due to a lack of relevant training and clinical practice experience, many healthcare providers have an insufficient scientific understanding of palliative care. In the clinical setting, they often struggle to appropriately guide patients and their families in selecting suitable treatment plans, thereby limiting the coverage and accessibility of palliative care [13]. Additional potential barriers to the practice of palliative care reflect attitudinal, informational, economic, social, and systemic obstacles, all of which are perceived by patients, physicians, and nurses [10]. The research investigated the knowledge, attitudes, and self-efficacy related to palliative care among Palestinian nurses working in intensive care units (ICUs). The results indicated that ICU nurses in the region demonstrated insufficient knowledge of palliative care and exhibited low self-efficacy in its administration [14]. These findings imply that gaps in knowledge and unfavorable attitudes among healthcare providers may constitute obstacles to the effective implementation of palliative care services. Another research has mentioned that perceptions of palliative care are shaped by factors such as age, gender, and department [15]. These findings suggest that various socio-demographic factors may significantly impact the level of knowledge regarding palliative care among nurses. As demonstrated in recent research, if doctors are unaware of or uncertain about the available palliative care options, they may influence or restrict referrals of terminally ill patients to palliative care facilities [16]. Research has shown that some individuals who have difficulty or discomfort accepting death may avoid discussing EOL issues with patients and their families. Referrals to palliative care may be perceived as a signal of loss of hope for the patient and their family members [17]. Furthermore, if referring physicians perceive or anticipate that patients and their families are unwilling to choose palliative care, the referral to palliative care facilities may be influenced or restricted [18]. Empirical findings provide evidence of a positive predictive relationship between healthcare providers' attitudes and behaviors towards graded diagnosis and treatment care.

However, negative attitudes towards graded diagnosis and treatment do not always translate into negative graded diagnosis and treatment behaviors. Protective or risk factors may influence those outcomes. Several studies have investigated potential mechanisms (such as mediators and moderators) between graded diagnosis and treatment behaviors and attitudes, predominantly focusing on patients or family members, with only a few examining healthcare providers in their research.

Self-efficacy is an individual's ability to execute and control the influences of their environment at a predetermined level [19]. As an internal protective factor, self-efficacy can potentially reduce the harmful consequences of stress on an individual's health [20]. Self-efficacy, as an intrinsic dynamic factor, can help individuals maintain a positive outlook and take action when facing challenges, enabling them to better cope with stress and difficulties. The Health Action Process Approach (HAPA) emphasizes self-efficacy's role across all behavior-change stages. In the motivational phase, self-efficacy affects behavioral intention, influencing an individual's readiness for specific health behaviors [21]. In the planning phase, it motivates individuals to formulate detailed action plans and attempt the planned actions. In the action phase, self-efficacy helps maintain actions despite challenges and coordinates resource use to sustain healthy behaviors. Overall, self-efficacy mediates attitudes, intentions, and actual behaviors throughout the health behavior change process. Additionally, research suggests a close relationship between self-efficacy and an individual's performance in executing designated tasks [22]. Empirical research findings indicate a positive correlation between attitudes toward graded diagnosis and treatment and self-efficacy.

Self-efficacy is also considered a key focal point and a protective factor for the well-being at work of healthcare providers. Numerous studies have identified a positive correlation between self-efficacy and the well-being at work of healthcare providers [23]. Well-being at work refers to an individual's subjective experience of contentment and happiness at work, reflecting a psychological state where self-fulfillment needs are met. However, surprisingly, the well-being of healthcare providers is declining, as evidenced by the high rates of professional burnout among physicians worldwide [24]. Research has shown that poor well-being at work among healthcare providers may lead to a decline in the quality of medical services, increased patient safety risks, and higher staff turnover rates [25]. Longitudinal studies have found that higher levels of compassion fatigue among healthcare providers are a risk factor for their well-being at work. Compassion fatigue refers to the emotional, physiological, and psychological exhaustion experienced by healthcare providers due to prolonged exposure to high-intensity work pressures [26]. Its consequences can affect nurses, organizations, and patients. It can occur suddenly, leading to emotional exhaustion, depression, helplessness, and a loss of well-being at work, especially among nurses [27]. In pediatric palliative care in the United States, it has been found that compassion fatigue can also impact personal well-being and professional effectiveness [28]. Therefore, gaining a deeper understanding of the potential mechanisms by which compassion fatigue impacts well-being at work is of paramount significance.

According to the stress-buffering model, the interaction between stressors (such as experiences of compassion fatigue) and self-efficacy can buffer the negative effects of stress (such as well-being at work) [29]. Self-efficacy refers to the crucial ability to execute and control one's environment at predetermined levels, influencing one's life [30]. It is closely related to an individual's psychosocial adaptation and has a direct impact on negative emotional experiences such as social anxiety, depression, worry, and hostility [31]. Individuals with high self-efficacy may be more inclined to actively process emotional information. This ability enables them to adopt more positive coping strategies when faced with experiences of compassion fatigue, leading to higher levels of well-being at work. However, some research also suggests that low levels of self-efficacy can serve as a protective factor against compassion fatigue [32]. Individuals with low self-efficacy may tend to protect their emotions and energy, avoiding excessive involvement in the emotional experiences of others, thus alleviating their empathic burden. As a stressor, compassion fatigue elicits different perceptions and cognitive evaluations among individuals with different levels of self-efficacy, leading to divergent levels of well-being at work [33]. Therefore, the personal implications vary. Consequently, self-efficacy may play a moderating role in the relationship between healthcare providers' compassion fatigue and well-being at work.

In conclusion, self-efficacy plays a significant role in healthcare providers' attitudes and behaviors towards graded diagnosis and treatment, compassion fatigue, and well-being at work. While the role of self-efficacy in nursing has been acknowledged, its mechanisms within the context of graded diagnosis and treatment remain unclear. Ignoring self-efficacy could lead to a lack of motivation among healthcare providers in clinical decision-making and work execution, thus affecting the quality and efficiency of healthcare services. Additionally, neglecting the potential impact of

self-efficacy on healthcare providers' psychological states and behaviors may exacerbate burnout, reduce job satisfaction, and weaken professional performance, ultimately impacting patient experience and the optimization of the healthcare system.

Therefore, this study aims to (1) investigate the relationship between healthcare providers' attitudes and behaviors toward palliative care graded diagnosis and treatment and the mediating role of self-efficacy, and (2) explore whether self-efficacy moderates the effect of compassion fatigue on well-being at work.

This study is based on the following hypotheses:

H1: Graded diagnosis and treatment attitude positively influences behavior and self-efficacy.

H2: Self-efficacy influences graded diagnosis and treatment behavior.

H3: Self-efficacy mediates the relationship between graded diagnosis and treatment attitude and behavior.

H4: Self-efficacy moderates the relationship between compassion fatigue and well-being at work.

## Methods

### Participants and procedure

A cross-sectional study was conducted on July 26, 2023, and completed on August 30, 2023, surveying 900 healthcare provider from two hospitals in Hubei Province, Wuhan (tertiary Grade A) and Huangshi (tertiary hospitals). The hospitals in Wuhan represent the palliative care graded diagnosis and treatment levels in provincial cities, while those in Huangshi reflect the graded diagnosis and treatment levels in regional cities. The number of beds in these two hospitals ranges from 1,800–3,300. The inclusion criteria were as follows: (a) participants were required to have a qualification certificate (nuresing or a physician license), and (b) willingness to participate in the study. According to the reference literature, when all the parameters are small and both the mediator and outcome are normal, approximately 371 observations are required to achieve a power of 0.8 for detecting a total indirect effect at the significance level of 0.05 [34]. A total of 900 survey forms were distributed. After data cleaning, 742 responses (validity rate of 77.6%) were deemed suitable for data analysis.

### Data collection methods

The researcher contacted the hospital, and secured approval from its nursing and physician departments. Then, face-to-face meetings were held with potential participants and the hospital director to explain the study's purpose, risks, and benefits. Trained research assistants from each hospital collected data following a standardised protocol and served as co-investigators. Data were gathered via an online survey platform (https://www.wjx.cn/). To ensure data integrity, each questionnaire could be submitted only once per IP address.

### Ethics approval and consent to participate

This study was approved by Zhongnan Hospital of Wuhan University Institutional Review Board (no. 2023041K). This study was registered with the China Clinical Trial Centre, registration number: ChiCTR2400085324. Written informed consent was voluntarily provided by all participants. The participants were free to withdraw at any time. Researchers assured participants that the collected data would be used solely for this research purpose. To safeguard individual identities, only numerical identifiers were used on questionnaires, eschewing any identifiable patient data. All participants were assigned unique identification numbers upon enrollment. These numbers were allocated in the sequence of questionnaire completion and stored separately from the informed consent documents.

### Measurement tools

**Demographic characteristics.** Demographic data of participants include: age, gender, educational background, professional title, marital status, department of work, and religious belief.

**Palliative care graded diagnosis and treatment knowledge, attitudes, and behaviors.** The Palliative Care Graded Diagnosis and Treatment Knowledge, Attitudes, and Behaviors questionnaire for healthcare providers was used to measure attitudes and behaviors towards palliative care graded diagnosis and treatment. This scale comprises three dimensions: knowledge, attitudes, and behaviors, totaling 46 items [35]. In the knowledge dimension, which contains 11 entries with single-choice answers, participants were primarily surveyed on their knowledge of palliative care graded diagnosis and treatment. The attitude dimension had 22 items, rated on a 5-point scale (1 = strongly disagree, 5 = strongly agree). Its total score ranged from 22 to 110, with higher scores indicating more positive attitudes towards palliative care graded diagnosis and treatment. The behavior dimension consists of 13 items, rated from 1(very rarely) to 5(always), and has a total score range of 13–65. Higher scores indicate more proactive engagement in palliative care graded diagnosis and treatment. The Cronbach's α coefficients for the attitude and behavior dimensions of this questionnaire are 0.959 and 0.924, respectively, with a total Cronbach's α of 0.937 in this study.

**General self-efficacy scale.** The General Self-Efficacy Scale (GSES), developed by Ralf Schwarzer, was used to measure nurses' self-efficacy [36]. This scale consists of 10 items, rated on a 4-point scale (1 = not at all true, 4 = exactly true). Total scores range from 10 to 40, with higher scores indicating a better positive expectation of ability. The Cronbach's alpha for this scale in this study was 0.948.

**Brief compassion fatigue scale.** The Brief Compassion Fatigue Scale, developed by Adams [37], was used to assess compassion fatigue among healthcare providers. It included two dimensions: professional burnout and secondary trauma, with a total of 13 items. Items were rated on a 10-point scale (1 = never, 10 = very frequently), with higher scores indicating greater severity. The scale has good validity and reliability, with a Cronbach's alpha of 0.888 in this study.

**Nurses' well-being at work scale.** The Nurses' Well-being at Work Scale, developed by Chen Ling [38], comprises five dimensions: welfare benefits, job values, interpersonal relationships, managers, and job characteristics, totaling 19 items. This scale uses a 6-point Likert scale, ranging from 1 ('strongly disagree') to 6 ('strongly agree'), with higher scores indicating higher levels of job satisfaction. The Cronbach's alpha for this scale in this study was 0.935.

### Statistical analysis

Descriptive analysis and generalized linear models were used to depict the distribution of demographic characteristics and attitudes and behaviors towards palliative care graded diagnosis and treatment, respectively. Only the attitude subscale was retained as a predictor: the knowledge dimension showed a severe ceiling effect and near-zero variance, rendering it unsuitable for regression, whereas the behaviour dimension served as the study's outcome variable and therefore could not be included as an independent or mediating factor. All inferential tests were thus performed on attitude scores. Mediation and moderated mediation models were conducted using the PROCESS macro for SPSS [39]. Using 5000 bootstrap samples, bias-corrected 95% confidence intervals (CIs) were calculated. Firstly, Model 4 was employed to test whether the relationship between palliative care graded diagnosis and treatment behavior and attitude is mediated by self-efficacy. If the 95% CI of the indirect effect (path a*b) does not include 0, it indicates a significant mediation effect. Next, Model 1 was utilized to test for moderated mediation, specifically whether self-efficacy moderates the direct and indirect effects of compassion fatigue on well-being at work. Similarly, if the 95% CI of the interaction effect does not include 0, a significant moderated mediation effect can be established. Following the recommended Johnson-Neyman technique, conditional effects and confidence bands were plotted [40]. All statistical analyses were conducted using SPSS 25.0. Statistical significance was defined as a two-tailed p-value < 0.05. Additionally, all models controlled for covariates (age, gender, marital status, and education level), and the study variables were standardized before analysis.

## Results

### Demographic and job-related characteristics

Table 1 presents the demographic and job-related characteristics of the participants. Among the 742 participants, there were 117 males (15.77%) and 625 females (84.23%); 597 individuals (80.46%) were married. Regarding educational attainment, 62.40% of participants held a bachelor's degree. The majority of participants had no religious affiliation (660, 88.95%). Additionally, 190 participants (25.61%) had received palliative care-related professional training.

### Mediation analysis

As shown in Table 2, the results of the mediation analysis indicate that the total effect (path c) of attitudes towards palliative care graded diagnosis and treatment on behavior was significant (B = 0.299, $p < 0.001$). The significant coefficients of paths a (B = 0.161, $p < 0.001$) and b (B = 0.647, $p < 0.001$) suggest a positive correlation between attitudes toward palliative care graded diagnosis and treatment and self-efficacy, and between self-efficacy and palliative care graded diagnosis and treatment behavior, respectively. As shown in Table 3, the estimated indirect effect (path a*b) of attitudes on behavior through self-efficacy was 0.104 ($p < 0.001$), with a bias-corrected bootstrap 95% confidence interval of 0.077 to 0.135, indicating a statistically significant indirect effect of attitudes on behavior. Additionally, the direct effect of attitudes on behavior (path c') was significant (B = 0.1948, $p < 0.001$). Therefore, self-efficacy mediated the relationship between attitudes and graded diagnosis and treatment behavior, with the mediation effect accounting for 34.81% of the total effect. The model is shown in Fig 1.

### Moderated mediation analysis

Table 4 reports the results of the moderated mediation analysis. Model 1 used well-being at work as the dependent variable and compassion fatigue and self-efficacy as independent variables ($p < 0.001$). Model 2 included compassion fatigue, self-efficacy, and the interaction term of compassion fatigue and self-efficacy as independent variables, indicating the presence of a moderation effect. From the table, it is evident that the regression coefficient of the interaction term (compassion fatigue * self-efficacy) is 0.011 ($p < 0.0001$), suggesting that self-efficacy significantly moderates the impact of compassion fatigue on well-being at work. Further analysis is needed to examine how self-efficacy moderates this relationship, i.e., how the relationship between compassion fatigue and well-being at work depends on self-efficacy. See Fig 1.

### Johnson-Neyman analysis

The results of the Johnson-Neyman analysis can be depicted using marginal effects or significance region plots, as shown in Fig 2. The Johnson-Neyman marginal effects or significance region plot demonstrates that the interaction between self-efficacy and compassion fatigue predicts well-being at work, with each predictor acting as a moderator. Fig 2 shows the region of significance when self-efficacy acts as a moderator. These lines represent the slope coefficients for the values of the moderator. The green area signifies the significance region, indicating the range of values for the moderator variable where the slope between the predictor variable and the outcome variable is significantly different from zero. The specific values ranged from 3 to 35. The shaded areas on either side of the lines represent the 95% confidence intervals around the slope terms. The vertical dashed line indicates the Johnson-Neyman value, which was the point where the confidence interval on the moderator no longer crosses zero, indicating that the slope is significantly different from zero at this point.

## Discussion

This study examined a moderated mediation model among healthcare providers in China. Below, we will discuss the key findings of this study in detail.

**Table 1. Demographic and work-related characteristics of the participants(N = 742).**

| | Variable | Category | N Percentage (%) |
|---|---|---|---|
| Demography characteristics | Gender | Male | 117 (15.77) |
| | | Female | 625 (84.23) |
| | Age(years) | ≤25 | 68 (9.16) |
| | | 26–30 | 215 (28.98) |
| | | 31–35 | 217 (29.25) |
| | | 36–40 | 93 (12.53) |
| | | ≥41 | 149 (20.08) |
| | Marital status | Currently married | 597 (80.46) |
| | | Never married | 134 (18.06) |
| | | Divorced/ Widowed | 11 (1.48) |
| | Education level | Secondary school and below | 16 (2.16) |
| | | College degree | 216 (29.11) |
| | | Bachelor degree | 463 (62.40) |
| | | Postgraduate and above | 47(6.33) |
| | Religious belief | Buddhist | 82 (11.05) |
| | | Without religious affiliation | 660 (88.95) |
| Work-related characteristics | Years of experience (years) | ≤10 | 421 (56.74) |
| | | 11-20 | 207 (27.90) |
| | | ≥21 | 114 (15.36) |
| | Work rank | Junior | 373 (50.0) |
| | | Intermediate | 297 (40.03) |
| | | Senior | 72 (9.7) |
| | Profession | Physicians | 154 (20.76) |
| | | Nurses | 578 (77.90) |
| | | Other medical personnel, technicians, etc. | 10 (1.35) |
| | Work Department | Internal ward | 403 (54.32) |
| | | Surgical ward | 199 (26.82) |
| | | Emergency intensive ward | 21 (2.83) |
| | | Other departments | 119 (16.04) |
| | Palliative care referral criteria training | Yes | 190 (25.61) |
| | | No | 552 (74.39) |
| | End-stage patient care experiences | Yes | 326 (43.94) |
| | | No | 416 (56.07) |
| | Palliative care referral experiences | Yes | 140 (18.87) |
| | | No | 602 (81.13) |
| | Organisation had a referral system for terminally ill patients | Yes | 267 (35.98) |
| | | No | 128 (17.25) |
| | | Unknown | 347 (46.77) |

Firstly, our findings indicated that a more positive attitude towards palliative care graded diagnosis and treatment among healthcare providers was positively associated with their actual palliative care graded diagnosis and treatment behavior (B = 0.299, p < 0.001), supporting our hypothesis 1. This finding underscores the close relationship between the attitudes and behaviors of healthcare providers towards palliative care graded diagnosis and treatment. The classical

**Table 2. Regression analysis of the relationship between the research variables in the model (n = 742).**

| Outcome | Variable | Significance of regression coefficients | | Fitness indicator | | |
|---|---|---|---|---|---|---|
| | | Beta | t | R | R² | F |
| Graded diagnosis and treatment behavior behaviortreatment behavior | Attitude | 0.299 | 12.099*** | 0.406 | 0.165 | 146.373 *** |
| Self-efficacy | Attitude | 0.161 | 11.400*** | 0.387 | 0.149 | 129.969 *** |
| Graded diagnosis and treatment behavior | Attitude | 0.195 | 7.823*** | 0.528 | 0.279 | 143.001*** |
| | Self-efficacy | 0.647 | 10.805*** | | | |

\* *p* < .05. \*\* *p* < .01. \*\*\* *p* < .001.

**Table 3. Moderated mediation analysis(n = 742).**

| Effect type | Effect | Boot se | LLCI | ULCI | Proportion(%) |
|---|---|---|---|---|---|
| Total effect | 0.299 | 0.025 | 0.250 | 0.347 | |
| Direct effect | 0.195 | 0.025 | 0.146 | 0.244 | |
| Indirect effect | 0.104 | 0.015 | 0.077 | 0.135 | 34.81 |

\* *p* < .05. \*\* *p* < .01. \*\*\* *p* < .001.

psychological attitude-behavior relationship model regards attitude as the basis of behavior, suggesting that attitudes directly influence behavior [41]. In mainland China, death has historically been regarded as a taboo, influenced by Confucian, Taoist, and Buddhist philosophies as well as superstitious beliefs [42]. This cultural context has engendered a pervasive avoidance of death-related topics, which in turn has fostered negative attitudes among physicians toward graded diagnosis and treatment behavior in healthcare providers. A study conducted among oncologists in Australia examined referral patterns to palliative care services. Among the 699 specialists surveyed, 48% reported that over 60% of their patients were referred to specialized palliative care services. The most frequently cited reasons for referral were unmet needs beyond the scope of oncology care. Conversely, the primary reasons for reluctance to refer included physicians' confidence in their ability to provide the required treatment and their belief that the quality of care delivered within the oncology setting was equivalent to that provided by palliative care institutions [43]. However, psychologists have noted that the relationship between individual attitudes and behavior is not entirely consistent and may require situational cues or 'channel factors' to influence [44]. In this study, the majority of patients receiving graded diagnosis and treatment were those with a life expectancy of less than 6 months, determining the need for graded diagnosis and treatment. However, when healthcare providers encounter such complex scenarios, their attitudes are more easily influenced, subsequently shaping their behavior. Especially in the context of our organization's status as a large tertiary hospital, the authority of large tertiary hospitals makes the words and actions of their doctors even more influential. In China's healthcare culture, large tertiary hospitals are often seen as benchmarks in the medical field, and doctors' professionalism and medical decisions are highly regarded and trusted by both peers and patients. Therefore, the attitudes and behaviors of these authoritative doctors not only affect their own practice of palliative care graded diagnosis and treatment but may also influence the attitudes and behaviors of other healthcare providers through the demonstration effect.

As widely recognized, self-efficacy plays a crucial role in promoting mental well-being and coping with stress [45]. By cultivating self-efficacy, individuals can alter their attitudes towards problems and cope with difficulties, thus reducing the occurrence of anxiety and depression [46]. This optimistic mindset not only enhances life satisfaction and happiness but also strengthens confidence and ability to tackle challenges. Conversely, individuals with an optimistic outlook are more likely to believe in their problem-solving abilities, further reinforcing their sense of self-efficacy. Our study findings also

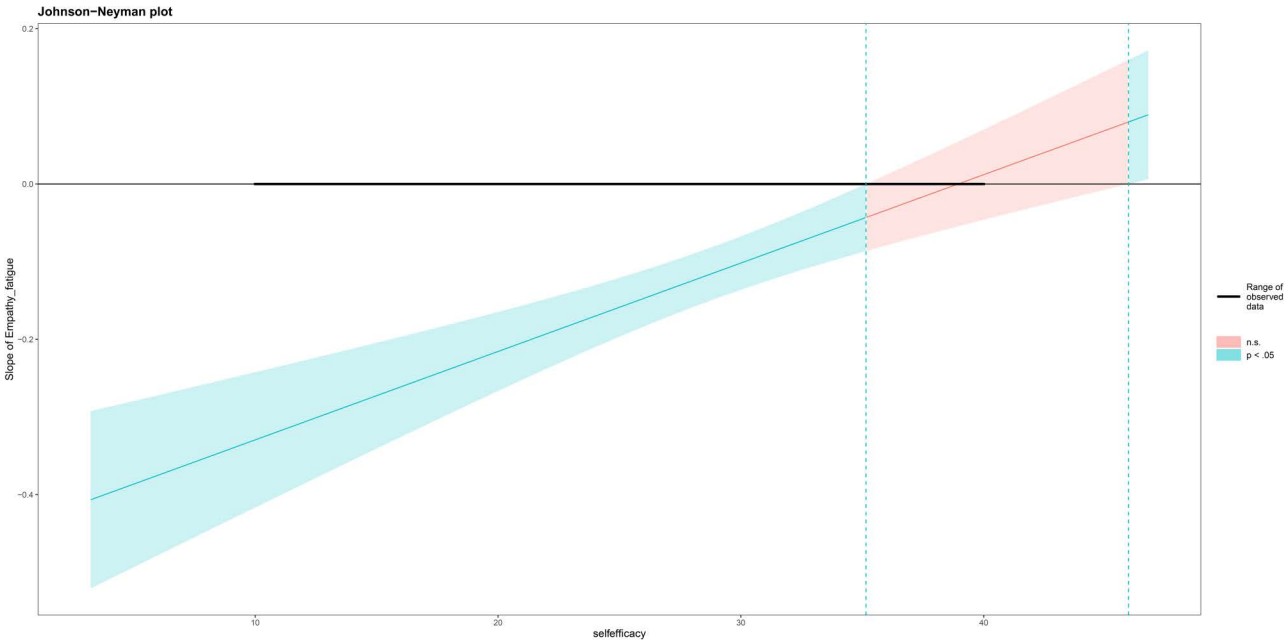

**Fig 1. Mediating moderation model.**

**Table 4. Regression analysis of compassion fatigue and self-efficacy on well-being at work (n = 742).**

|  | Model 1 well-being at work | Model 2 well-being at work |
|---|---|---|
| (Intercept) | 81.191*** | 81.203*** |
|  | (0.599) | (0.489) |
| Compassion fatigue | −0.116*** | −0.143*** |
|  | (0.021) | (0.018) |
| Self-efficacy |  | 1.363*** |
|  |  | (0.072) |
| Compassion fatigue × self-efficacy |  | 0.011*** |
|  |  | (0.002) |
| $R^2$ | 0.039 | 0.362 |
| Adj. $R^2$ | 0.037 | 0.359 |
| Num. obs. | 742 | 742 |

*Note*. Unstandardized regression coefficients are displayed, with standard errors in parentheses.

* $p < .05$. ** $p < .01$. *** $p < .001$.

indicate a positive correlation between healthcare providers' favorable attitudes towards graded diagnosis and treatment and their self-efficacy (B = 0.161, p < 0.001), thus further supporting our hypothesis 1. Previous research has predominantly focused on the educational domain, examining the relationship between students' self-efficacy and their learning attitudes [47]. Studies have reported positive effects of academic self-efficacy on nursing students' learning attitudes in nursing education [48]. However, research focusing on healthcare providers, particularly in the context of palliative care graded diagnosis and treatment is limited. Therefore, our study, by highlighting the importance of self-efficacy in this

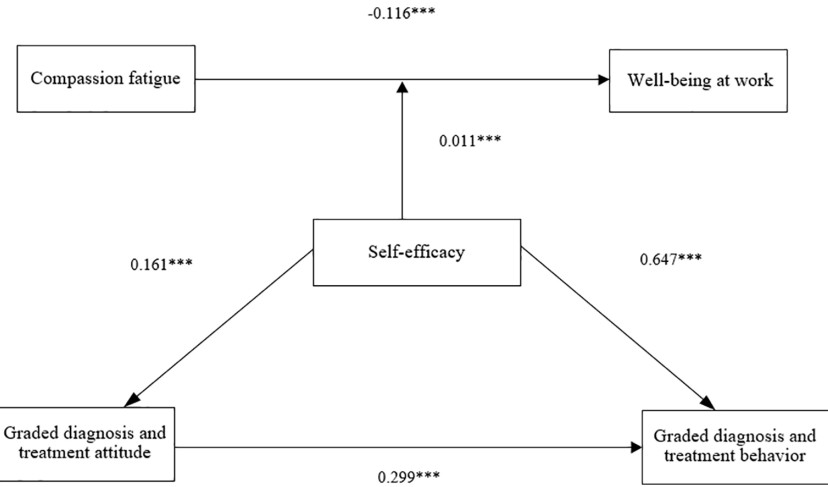

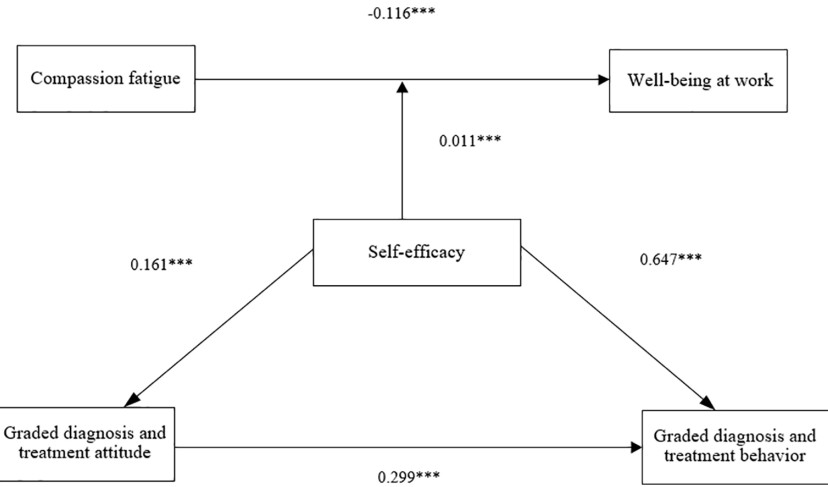

**Fig 2. Johnson-Neyman moderating effect graph.**

process, offers effective pathways and strategies to enhance healthcare providers' active engagement in graded diagnosis and treatment.

The results show that self-efficacy, as a protective factor, positively influences healthcare providers' graded diagnosis and treatment behavior (B = 0.647, p < 0.001), thus confirming Hypothesis 2. Healthcare providers who effectively utilize protective factors are more likely to proactively report greater engagement in graded diagnosis and treatment activities. This enhances their confidence and motivation, fostering proactive behavior, personal growth, reflective self-improvement, as well as exploration of life goals and self-enhancement. According to self-efficacy theory, an individual's confidence level directly impacts their approach to task execution and performance outcomes [49]. This mechanism aligns with prior evidence that individuals with higher self-efficacy are more likely to adopt proactive coping strategies under stress [50]. The Health Belief Model suggests that increases in self-efficacy lead to changes in behavior over time [51]. For example, research has confirmed that self-efficacy is closely associated with the adoption and engagement in various health behaviors, such as smoking cessation. Compared to daily smokers, non-daily smokers exhibit higher levels of self-efficacy [52]. However, research on the impact of self-efficacy on the graded diagnosis and treatment behavior of healthcare providers is currently limited in quantity. Thus, this study offers significant insights and contributes a new perspective to this area of knowledge. By identifying the relationship between healthcare providers' self-efficacy and graded diagnosis and treatment behavior, we can explore methods to increase their self-efficacy, thus influencing their behavior in graded diagnosis and treatment. This offers beneficial insights for further improving healthcare services, potentially helping healthcare providers to be more effective in engaging in graded diagnosis and treatment, enhancing patient outcomes, and promoting advances in the overall healthcare system.

Interestingly, the results of this study indicate that self-efficacy mediates the relationship between healthcare providers' attitudes and behaviors, with the mediation effect accounting for 34.81% of the total effect, thus supporting Hypothesis 3. These findings suggest that healthcare providers' self-efficacy helps adjust their attitudes toward palliative care graded diagnosis and treatment and increases their willingness and initiative to implement graded diagnosis and treatment. In a study conducted at 10 compulsory isolation drug rehabilitation centers in Zhejiang Province, 1,197 Substance use disorder (SUDs) patients were recruited [53]. The findings indicated a positive effect of attitudinal behavior on behavioral intention (β = 0.02), while self-efficacy did not emerge as a mediator. In contrast, self-efficacy appeared as a mediating variable in this study, with the healthcare providers population and showed a large effect size (34.81%). Healthcare providers may derive their self-efficacy from their

education, training, and work experience, which enhances their confidence in their abilities. Higher self-efficacy is essential for healthcare providers to overcome the challenges and pressures of their work, making it a crucial factor in forming their behavioral intentions. Different behavioral intentions may require varying levels of self-efficacy to support them; for instance, healthcare providers' behavioral intentions may relate to the provision of care and interventions, whereas SUD patients. However, unlike this study, it did not explore the mediating role of self-efficacy between the attitudes and behavior of healthcare providers. The present study, by calculating the effect size of the mediating effect (34.81%), provides a quantitative basis for understanding the actual impact of self-efficacy. It further reveals how self-efficacy can change healthcare providers' behaviors by influencing their attitudes, which is important and practical for optimizing the practice of graded diagnosis and treatment care.

Lastly, but equally important, a significant finding of this study is the moderating role of self-efficacy in the relationship between compassion fatigue and well-being at work, partially supporting Hypothesis 4. This finding suggests that healthcare providers' self-efficacy can influence their well-being at work by impacting their compassion fatigue, a negative emotion. Healthcare providers with high self-efficacy handle EOL care tasks with confidence and competence [54]. They can cope with emotional challenges, establish positive relationships, and alleviate fatigue. The healthcare providers included in this study were generally exposed to an occupational environment characterized by strict management standards and high work intensity, which could easily overload and significantly impact their well-being and empathic ability. This overload could further exacerbate their empathic fatigue, making it difficult for them to cope with their patients. Additionally, the support system and overall working environment of the hospital play a crucial role in the self-efficacy of healthcare providers. A survey conducted among 992 nurses in central China revealed that nurses with higher self-efficacy tend to view compassionate stress as an opportunity rather than a threat in their daily work [55]. They believe in their ability to overcome stress rather than avoid it, leading to lower levels of professional burnout. This experience brings satisfaction and a sense of accomplishment as they can provide care and support to patients during critical moments, experience the meaning of their work, and establish deep connections with both their team and patients. In contrast, a decrease in healthcare providers' self-efficacy exacerbates work fatigue, thereby reducing their well-being at work [56]. In such situations, they may feel increasingly powerless to face the challenges and pressures of work, leading to physical and mental exhaustion, thereby impacting well-being at work and overall quality of life. Similar to this study, a study conducted on 712 hotline psychological counselors from the Psychological Health Service Platform of Central China Normal University found that self-efficacy independently and continuously moderated the relationship between compassion fatigue and self-oriented empathy [32]. Self-oriented empathy is the phenomenon of projecting one's own emotions onto others or situations, often stemming from resonance with others or situations [57]. For example, projecting emotions to project goals may enhance team belongingness and satisfaction, thus boosting well-being at work. However, there is currently no research confirming whether self-efficacy can independently and continuously mediate the relationship between compassion fatigue and well-being at work. In conclusion, this finding provides new insights for clinical practice, suggesting that by providing training, support, and psychological health resources to enhance healthcare providers' confidence and abilities, there is potential to improve their well-being at work.

## Limitations

Our study has several limitations. First, we used a cross-sectional design to assess hypotheses and were able to identify causal relationships for the mediating variables, but were unable to infer dynamic changes in these relationships over time. Secondly, we used four self-report tools to collect data, which may lead to response or social desirability biases, thus affecting participants' responses. We acknowledge that the results obtained from the self-reported measures should be interpreted with caution due to the potential limitations inherent in such data collection methods. Our sample was limited to healthcare providers from two hospitals in Hubei Province, China, which may result in insufficient sample representativeness, and potential confounding variables were not accounted for in the study. Thus, the generalizability of our study is limited, and its conclusions should be cautiously interpreted.

## Conclusion

Despite some limitations, this study holds significant theoretical and practical implications. It underscores the pivotal role of healthcare providers in palliative care quality, emphasizing the importance of their attitudes and behaviors. Self-efficacy has been identified as a key determinant influencing their engagement in graded diagnosis and treatment and serves as a mediator between these factors. Enhancing professionals' self-efficacy can strengthen their willingness and motivation to participate in graded diagnosis and treatment, thus improving the quality of palliative care. Furthermore, the study confirms the moderating effect of compassion fatigue on well-being at work, providing theoretical support and practical guidance for future clinical practices. Future research should explore training programs aimed at boosting healthcare providers' self-efficacy to advance palliative care and elevate healthcare service quality. Additionally, future studies are encouraged to integrate observational methods (e.g., longitudinal interviews, retrospective cohorts) to complement self-report limitations and account for individual variability, enhancing research validity.

## Implications

This study highlights the critical role of healthcare providers in grading diagnosis and treatment of palliative care and underscores the importance of enhancing self-efficacy to improve the quality of palliative treatment. In clinical practice, a well-structured graded diagnosis and treatment system helps allocate palliative care resources effectively, preventing resource waste and excessive medical interventions. Large hospitals leverage their technological advantages to provide high-level treatments, while smaller hospitals and community healthcare facilities offer more practical care, with ample bed capacity and convenient locations, reducing patient burdens and improving treatment efficiency.

Self-efficacy emerged as a key determinant and mediator of providers' graded-care participation, guiding future training design.Translating psychosocial evidence into scalable action, through targeted training and interventions, boosting their confidence and decision-making abilities can effectively encourage their active participation in complex situations and improve the quality of graded diagnosis and treatment.

Hospitals should offer psychological support to alleviate stress and prevent burnout, thereby enhancing self-efficacy. Only by improving self-efficacy can healthcare providers execute tasks more decisively and effectively in palliative care graded diagnosis and treatment, driving the rational allocation of resources and the integration of palliative care principles. Moreover, hospitals can optimize resource allocation, improve healthcare quality, and ultimately achieve the dual goals of enhancing patient experience and increasing healthcare providers' job satisfaction by promoting policies, sharing best practices, and offering economic incentives.

## Supporting information

**S1 Data. Supplementary Information Data.**
(XLSX)

## Author contributions

**Conceptualization:** Lixiang Liu.

**Data curation:** Lixiang Liu, Meidi Xiong, Fang Hong, Yiting Zhang, Chunhua Zhang.

**Formal analysis:** Lixiang Liu, Yiting Zhang.

**Funding acquisition:** Chunhua Zhang.

**Methodology:** Lixiang Liu.

**Project administration:** Chunhua Zhang.

**Resources:** Fang Hong.

**Software:** Meidi Xiong, Yiting Zhang.

**Supervision:** Fang Hong, Chunhua Zhang.

**Validation:** Meidi Xiong.

**Visualization:** Fang Hong.

**Writing – original draft:** Lixiang Liu, Meidi Xiong.

**Writing – review & editing:** Chunhua Zhang.

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
