## [Decision Letter · Decision Letter 0]

29 Dec 2024

Dear Dr. Zhang,

Thank you for submitting your manuscript to PLOS ONE. After careful consideration, we feel that it has merit but does not fully meet PLOS ONE’s publication criteria as it currently stands. Therefore, we invite you to submit a revised version of the manuscript that addresses the points raised during the review process.

We look forward to receiving your revised manuscript.

Kind regards,

Amal Diab Ghanem Atalla, ph.D

Academic Editor

PLOS ONE

Journal Requirements:

This work was Supported by Special Research Cultivation Fund for Clinical Nursing of Wuhan University, Project LCHL202327. 

3. "In the online submission form, you indicated that the data that support the findings of this study are available on request from the corresponding author. The data are not publicly available due to privacy orethical restrictions.

Additional Editor Comments :

Dear Author,

Thank you for submitting your manuscript titled "[Title of Manuscript]." We have received comprehensive feedback from the reviewers, which provides constructive insights to further improve the quality and clarity of your work. We kindly request you to carefully revise the manuscript in accordance with their comments and suggestions. Please ensure that all points raised by the reviewers are addressed thoroughly, and where necessary, provide a detailed explanation for any comments that cannot be fully implemented. Once revised, please include a response letter summarizing the changes made and how each reviewer’s comment has been addressed. We look forward to receiving your revised submission.

Best regards,

[Amal Diab]

Reviewers' comments:

Reviewer's Responses to Questions

**Comments to the Author**

1. Is the manuscript technically sound, and do the data support the conclusions?

Reviewer #1: Yes

Reviewer #2: Yes

Reviewer #3: Yes

2. Has the statistical analysis been performed appropriately and rigorously?

Reviewer #1: Yes

Reviewer #2: Yes

Reviewer #3: Yes

3. Have the authors made all data underlying the findings in their manuscript fully available?

Reviewer #1: Yes

Reviewer #2: Yes

Reviewer #3: Yes

4. Is the manuscript presented in an intelligible fashion and written in standard English?

Reviewer #1: Yes

Reviewer #2: Yes

Reviewer #3: Yes

Reviewer #1: thank you for your effort to modify your article in form that improvment make it very useful in increasing body of knowlege

thanks again and keep going , but i finnaly recommend minir english revision to improve rediability of your manuscript

Reviewer #2: Thanks for inviting me for reviewing this topic Please consider the following:

- clarify more practical implications for nursing in abstract and implications section

- introduction: the significance of the study need to be highlighted and more clarified

- clarify more the rationale for selecting such settings with detailed description.

- reliability and pilot study clarify whether it is from the same study hospital or different one??

- Implications for practice need to be modified to more applicable actions

- clarify more strengths and limitations of the study

Reviewer #3: One point should be considered for using more updated references 2023-2024 in the literature. The sampling should be discussed in detail with references. The discussion should link all variables of the study with literature and results. More variables should be included in the study to make it stronger. some paragraph need to add references. You need to justify to make it clear to generalize your results. Data Collection need more details.

**Do you want your identity to be public for this peer review?** For information about this choice, including consent withdrawal, please see our Privacy Policy

Reviewer #1: **Yes: ** Mostafa shaban

Reviewer #2: No

Reviewer #3: No

---

## [Author Response · Author response to Decision Letter 1]

17 Feb 2025

Reviewer #1: thank you for your effort to modify your article in form that improvment make it very useful in increasing body of knowlege thanks again and keep going , but i finnaly recommend minir english revision to improve rediability of your manuscript

Response: Thank you very much for your kind words and valuable feedback. I have completed the minor English revisions as you suggested to further enhance the readability of the paper. Your feedback has been incredibly helpful in refining the manuscript, and I am grateful for your support. Thank you once again for your guidance!

Reviewer #2: Thanks for inviting me for reviewing this topic Please consider the following:

1.clarify more practical implications for nursing in abstract and implications section

Response: We would like to express our sincere gratitude to the reviewer for the valuable feedback, particularly the suggestion to further elaborate on the practical significance for nursing in both the abstract and implications sections. Your comments have greatly assisted us in clearly articulating the practical applications of this study, especially in how enhancing healthcare professionals' self-efficacy can optimize the grading of palliative care and improve nursing quality. Your insights have significantly contributed to refining and deepening the discussion of the study's significance, playing a crucial role in the improvement of the research. We sincerely appreciate your thoughtful input.

2. introduction: the significance of the study need to be highlighted and more clarified

Response: Thank you for your valuable feedback. I appreciate your suggestion regarding the need to further highlight and clarify the significance of the study. In response, I have added additional clarification in the final paragraph of the introduction.

The revised content is as follows:

“In conclusion, self-efficacy plays a significant role in healthcare professionals' attitudes and behaviors towards graded treatment, compassion fatigue, and well-being at work. While the role of self-efficacy in nursing has been acknowledged, its mechanisms within the context of graded treatment remain unclear. Ignoring self-efficacy could lead to a lack of motivation among healthcare professionals in clinical decision-making and work execution, thus affecting the quality and efficiency of healthcare services. Additionally, neglecting the potential impact of self-efficacy on healthcare professionals' psychological states and behaviors may exacerbate burnout, reduce job satisfaction, and weaken professional performance, ultimately impacting patient experience and the optimization of the healthcare system.

Therefore, this study aims to investigate whether self-efficacy mediates the relationship between healthcare professionals' attitudes and behaviors towards graded treatment and to evaluate a moderated mediation model. We hypothesize that self-efficacy may mediate the relationship between attitudes and behaviors, and also moderate the indirect effect of compassion fatigue on well-being at work.”

3. clarify more the rationale for selecting such settings with detailed description.

Response: Thank you for your valuable feedback. I appreciate your suggestion regarding the need to clarify the rationale for selecting the study settings. In response, I would like to explain that the hospital in Wuhan was chosen due to its advantageous geographical location, large scale, and extensive case pool, which made sample collection convenient. Additionally, as I work in this hospital, it provided a more accessible and practical setting for conducting the study.

On the other hand, the tertiary hospital in Huangshi, a non-provincial capital city, was included to ensure the study covered healthcare institutions in smaller cities, which offers a broader regional perspective. This approach enhances the study’s applicability to a wider range of healthcare settings. To collect data, I employed convenience sampling, which ensured the feasibility of gathering the sample while allowing for a comprehensive understanding of the healthcare services in different regions.

4. reliability and pilot study clarify whether it is from the same study hospital or different one?

Response: Thank you for your valuable feedback. Regarding your inquiry about reliability and the pilot study, I would like to clarify that the pilot study was conducted at both the Wuhan and Huangshi hospitals, which were the same hospitals used for the main data collection. If you believe it is necessary to explicitly highlight this in the article, I will promptly make the necessary adjustments and provide further clarification.

5.Implications for practice need to be modified to more applicable actions

Response: Thank you for your feedback. I have made revisions based on your suggestions. Below is the updated section on implications for practice.

The revised content is as follows:

“In conclusion, this finding provides valuable insights for clinical practice. Enhancing healthcare professionals' self-efficacy is key to successfully implementing tiered diagnosis and treatment. Through targeted training and interventions, boosting their confidence and decision-making abilities can effectively encourage their active participation in complex situations and improve the quality of tiered care. Hospitals should offer psychological support to alleviate stress and prevent burnout, thereby enhancing self-efficacy. Only by improving self-efficacy can healthcare professionals execute tasks more decisively and effectively in tiered care and palliative care, driving the rational allocation of resources and the integration of palliative care principles. Moreover, hospitals can optimize resource allocation, improve healthcare quality, and ultimately achieve the dual goals of enhancing patient experience and increasing healthcare professionals' job satisfaction by promoting policies, sharing best practices, and offering economic incentives.”

6.clarify more strengths and limitations of the study

Response: Thank you for your suggestion. I have expanded the content to provide a more comprehensive explanation of the study's strengths and limitations. The revised section addresses considerations such as potential confounding variables and sample representativeness, offering a more balanced perspective.

Reviewer #3: One point should be considered for using more updated references 2023-2024 in the literature.

1.The sampling should be discussed in detail with references.

Response: Thank you for your valuable feedback. According to the reference literature,when all the parameters are small and both the mediator and outcome are normal, approximately 371 observations are required to achieve a power of 0.8 for detecting a total indirect effect at the significance level of 0.05.

2.The discussion should link all variables of the study with literature and results.

Response: Thank you for your valuable feedback. I have revised the discussion section to better link all the variables in the study with both the relevant literature and the results. In the updated version, I have provided a more detailed analysis of each variable, comparing the findings with existing research to highlight similarities and differences.I appreciate your insightful suggestion, and it has been incorporated into the revision. Thank you once again for your guidance!

3.More variables should be included in the study to make it stronger.

Response: Thank you very much for your thoughtful suggestion. Currently, the study includes five key variables, and this selection was influenced by previous guidance I received, where it was emphasized that variables should be chosen based on positive indicators reported in the literature, rather than aiming to include as many variables as possible. The focus was placed on selecting the most relevant and meaningful variables for the research. That said, I deeply appreciate your input, and I will definitely keep your advice in mind as the study progresses. In the future, I hope to expand the scope of the study and consider adding more variables as the research develops. Thank you again for your valuable guidance!

4. some paragraph need to add references.

Response: Thank you for your valuable feedback. I have added references where necessary.

5.You need to justify to make it clear to generalize your results.

Response: Thank you again for your insightful suggestion. I have made the necessary revisions to the manuscript, where I have provided a clear justification for the generalizability of the study results. I have explained the scope and limitations of the study, along with the steps taken to ensure the validity and reliability of the findings. This should help clarify how the results can be applied to similar populations or settings. I appreciate your feedback, and I believe these changes strengthen the manuscript. Thank you once again for your valuable guidance!

6. Data Collection need more details.

Response: Thank you for your suggestion. Data were collected using an online survey platform generated from a dedicated URL link ( https://www.wjx.cn/). Survey WeChat QR code (a Chinese instant messaging app ) images were disseminated to the nursing program directors of the nine institutions, who also served as co-investigators.

---

## [Decision Letter · Decision Letter 1]

1 Apr 2025

Dear Dr. Zhang,

Thank you for submitting your manuscript to PLOS ONE. After careful consideration, we feel that it has merit but does not fully meet PLOS ONE’s publication criteria as it currently stands. Therefore, we invite you to submit a revised version of the manuscript that addresses the points raised during the review process.

**ACADEMIC EDITOR: **

We look forward to receiving your revised manuscript.

Kind regards,

Professor Amal Diab Ghanem Atalla

Academic Editor

PLOS ONE

Journal Requirements:

Additional Editor Comments:

Dear Author

Following our review of your manuscript, we have identified minor revisions that will enhance its clarity and readability. These revisions primarily involve grammar editing, proofreading, and slight refinements to improve sentence structure, coherence, and overall flow. While these changes do not affect the core findings or conclusions of your study, they will ensure a polished and professional presentation.

We kindly request that you review and incorporate these revisions before resubmitting the manuscript. Should you need any clarification, please do not hesitate to reach out.

Thank you for your efforts, and we look forward to receiving the revised version.

Best regards

Reviewers' comments:

Reviewer's Responses to Questions

**Comments to the Author**

Reviewer #2: (No Response)

Reviewer #4: All comments have been addressed

2. Is the manuscript technically sound, and do the data support the conclusions?

Reviewer #2: Yes

Reviewer #4: Partly

3. Has the statistical analysis been performed appropriately and rigorously?

Reviewer #2: Yes

Reviewer #4: Yes

4. Have the authors made all data underlying the findings in their manuscript fully available?

Reviewer #2: Yes

Reviewer #4: Yes

5. Is the manuscript presented in an intelligible fashion and written in standard English?

Reviewer #2: Yes

Reviewer #4: Yes

Reviewer #2: Areas for Improvement:

1. Abstract & Introduction

The abstract lacks clarity in describing key findings in a concise manner.

Recommendation: Summarize the major statistical results more succinctly while emphasizing the study’s novelty.

The introduction provides a broad background but could benefit from a stronger justification for why self-efficacy was chosen as a mediating/moderating factor.

Recommendation: Strengthen the theoretical underpinning for why self-efficacy is expected to influence both attitudes and behavior in palliative care.

2. Methodology

The study relies entirely on self-reported measures, which can introduce bias.

Recommendation: Acknowledge this limitation more explicitly and suggest potential complementary methods, such as observational studies or supervisor evaluations.

The justification for using SPSS PROCESS for mediation analysis is not clearly articulated.

Recommendation: Briefly explain why this statistical approach was chosen over alternatives like Structural Equation Modeling (SEM).

3. Results & Interpretation

The mediation effect (34.81%) is moderate but not overwhelmingly strong. This nuance is not discussed.

Recommendation: Interpret the mediation effect in the context of previous studies—how does this compare with similar research?

While the moderation model is statistically significant, the practical significance (effect size) is not explicitly discussed.

Recommendation: Provide effect size measures to contextualize the practical importance of the findings.

4. Discussion & Implications

The discussion restates results but lacks deeper engagement with alternative explanations.

Recommendation: Consider potential confounders, such as institutional culture or workload, that may have influenced results.

The discussion does not address cultural factors in China that may impact attitudes toward palliative care.

Recommendation: Include a brief discussion on cultural attitudes towards end-of-life care and whether findings might differ in other healthcare systems.

5. Limitations & Future Directions

The limitations section is well-structured but could mention response bias more explicitly.

Recommendation: Suggest future studies incorporating mixed-methods approaches to mitigate self-reporting bias.

Final Recommendation:

Accept with Minor Revisions

Improve clarity in abstract and introduction.

Provide deeper discussion on effect sizes and alternative explanations.

Address cultural considerations and response bias more explicitly.

This study has strong potential and makes a meaningful contribution to palliative care research. With these refinements, it would be even stronger.

Reviewer #4: Dear authors, thank you for trying to deepen the knowledge within your topic, I have taken great interest in reading your work.

- Please ensure that your manuscript meets all requirements

- The title is very critical and important, additionally it covers an extra point of research not covered in previous research.

- Introduction; it well constructed please focus on paraphrasing

- Methodology: Please mention more specifically why select these two hospitals

- Data collection process very short give more clarification about actual process regarding the duration to fill every sheet, additionally how you decrease the bias during data collection process, Moreover, more clarification about the role of co-investigators

-The Palliative Care Graded Diagnosis and Treatment Knowledge, Attitudes, and Practices tool please tension about this paragraph “In the knowledge dimension, participants were primarily surveyed on their knowledge of palliative care graded diagnosis and treatment, rated on a 5-point scale (1 = strongly disagree, 5 = strongly agree). Higher scores indicate more positive attitudes towards palliative care graded diagnosis and treatment. The behavior dimension consists of 13 items, rated from 1(very rarely) to 5(always), with higher scores indicating more proactive engagement in palliative care graded diagnosis and treatment. “ please more clarification about the three dimensions in this tool about how many items in each dimension because you mention knowledge dimension and you mention this dimension used to assess attitude and you didn’t mention the attitude dimension itself and how many items????.

- In the General Self-Efficacy Scale (GSES), developed by Ralf Schwarzer, was used to measure nurses’ self-efficacy , you didn’t mention the Ralf Schwarzer article in the reference section

- What about pilot sample ?

- This manuscript requires additional editing and proofreading to correct detected errors

- Please add DOI in some references

I wish you successfully continued work with your manuscript!

**Do you want your identity to be public for this peer review?** For information about this choice, including consent withdrawal, please see our Privacy Policy

Reviewer #2: **Yes: ** Samia Roshdy Soliman Osman

Reviewer #4: No

---

## [Author Response · Author response to Decision Letter 2]

14 May 2025

Reviewer #2: Areas for Improvement:

1. Abstract & Introduction

1.1 The abstract lacks clarity in describing key findings in a concise manner.

Recommendation: Summarize the major statistical results more succinctly while emphasizing the study’s novelty.

Response: Thank you very much for your kind words and valuable feedback. The content after our revisions is as follows: While prior research has acknowledged the role of self-efficacy in nursing, its specific mechanisms within graded treatment contexts remain underexplored. Our study investigates how self-efficacy bridges healthcare professionals' attitudes and behaviors towards graded treatment with compassion fatigue and workplace well-being.

Health care professionals' attitudes towards grading treatment predicted an increase in self-efficacy (β = 0.161, p < 0.001), which subsequently led to improved grading behavior in treatment (β = 0.647, p < 0.001). Self-efficacy mediated 34.81% of the effect of attitudes on grading behavior.

1.2 The introduction provides a broad background but could benefit from a stronger justification for why self-efficacy was chosen as a mediating/moderating factor.

Recommendation: Strengthen the theoretical underpinning for why self-efficacy is expected to influence both attitudes and behavior in palliative care.

Response: Thank you for your valuable feedback. The Health Action Process Approach (HAPA) emphasizes self-efficacy's role across all behavior - change stages. In the motivational phase, self-efficacy affects behavioral intention, influencing an individual's readiness for specific health behaviors[19]. In the planning phase, it motivates individuals to formulate detailed action plans and attempt the planned actions. In the action phase, self-efficacy helps maintain actions despite challenges and coordinates resource use to sustain healthy behaviors. Overall, self-efficacy mediates attitudes, intentions, and actual behaviors throughout the health behavior change process.

2. Methodology

2.1 The study relies entirely on self-reported measures, which can introduce bias.

Recommendation: Acknowledge this limitation more explicitly and suggest potential complementary methods, such as observational studies or supervisor evaluations.

Response: Thanks to the editor's suggestion, we've made the following changes:

Limitations We acknowledge that the results obtained from the self-reported measures should be interpreted with caution due to the potential limitations inherent in such data collection methods.

Conclusion Future studies should integrate observational methods (e.g., longitudinal interviews, retrospective cohorts) to complement self-report limitations and account for individual variability, enhancing research validity.

2.2 The justification for using SPSS PROCESS for mediation analysis is not clearly articulated. Recommendation: Briefly explain why this statistical approach was chosen over alternatives like Structural Equation Modeling (SEM).

Response: Thank you for your valuable feedback. We highly value your comments and have carefully reviewed the points raised. We’d like to clarify our choice of using the SPSS PROCESS macro for mediation analysis. Given the relatively simple mediation model in this study—focusing on a single mediator between the independent and dependent variables—SPSS PROCESS was deemed appropriate. It efficiently tests mediation effects and offers clear statistical results. Moreover, its seamless integration with SPSS makes it user-friendly, streamlining the analysis and interpretation process. Additionally, it provides various methods for mediation analysis, such as Sobel and Bootstrap, enhancing analytical flexibility. However, we acknowledge the usefulness of Structural Equation Modeling (SEM) and will consider it for future studies with larger sample sizes and more complex models.

3. Results & Interpretation

3.1 The mediation effect (34.81%) is moderate but not overwhelmingly strong. This nuance is not discussed.

Recommendation: Interpret the mediation effect in the context of previous studies—how does this compare with similar research?

Response: Thank you so much for your insightful feedback. In a study conducted at 10 compulsory isolation drug rehabilitation centers in Zhejiang Province, 1,197 Substance use disorders (SUDs) patients were recruited. The findings indicated a positive effect of attitudinal behavior on behavioral intention (β = 0.02), while self-efficacy did not emerge as a mediator. In contrast, self-efficacy appeared as a mediating variable in this study, with the nurse population and showed a large effect size (34.81%). Nurses, as professionals, may derive their self-efficacy from their education, training, and work experience, which enhances their confidence in their abilities. Higher self-efficacy is essential for nurses to overcome the challenges and pressures of their work, making it a crucial factor in forming their behavioral intentions. Different behavioral intentions may require varying levels of self-efficacy to support them; for instance, nurses' behavioral intentions may relate to the provision of care and interventions, whereas SUD patients'.

3.2 While the moderation model is statistically significant, the practical significance (effect size) is not explicitly discussed.

Recommendation: Provide effect size measures to contextualize the practical importance of the findings.

Response: Thank you for your valuable feedback. I have now explicitly discussed the practical significance of the findings by providing effect size measures. In the revised section, I have included a detailed analysis of the effect size, which shows that the mediation effect of self-efficacy on the relationship between healthcare providers' attitudes and their tiered diagnosis and treatment behaviors accounts for 34.81% of the total effect. This effect size underscores the practical importance of self-efficacy in shaping healthcare providers' attitudes and behaviors, thereby supporting Hypothesis 3. I believe this addition provides a clearer understanding of the practical implications of the study results.

4. Discussion & Implications

4.1 The discussion restates results but lacks deeper engagement with alternative explanations.

Recommendation: Consider potential confounders, such as institutional culture or workload, that may have influenced results.

Response: Thank you for your valuable feedback. We have carefully considered your suggestion regarding the need for deeper engagement with alternative explanations in the discussion section. In response, we have revised the discussion to include a more thorough examination of potential confounders that may have influenced our results. Specifically, we have now incorporated an analysis of institutional culture and workload as possible factors that could have affected the outcomes observed in our study. We believe that this enhanced discussion provides a more comprehensive understanding of our findings and acknowledges the complexity of the factors at play.

4.2 The discussion does not address cultural factors in China that may impact attitudes toward palliative care. Recommendation: Include a brief discussion on cultural attitudes towards end-of-life care and whether findings might differ in other healthcare systems.

Response: Thank you so much for your insightful feedback. In mainland China, death has historically been regarded as a taboo, influenced by Confucian, Taoist, and Buddhist philosophies as well as superstitious beliefs[40]. This cultural context has engendered a pervasive avoidance of death-related topics, which in turn has fostered negative attitudes among physicians toward triage behavior in healthcare professionals.

5. Limitations & Future Directions

The limitations section is well-structured but could mention response bias more explicitly. Recommendation: Suggest future studies incorporating mixed-methods approaches to mitigate self-reporting bias.

Response: Thank you for your valuable feedback. We acknowledge that the results obtained from the self-reported measures should be interpreted with caution due to the potential limitations inherent in such data collection methods. Future studies should integrate observational methods (e.g., longitudinal interviews, retrospective cohorts) to complement self-report limitations and account for individual variability, enhancing research validity.

Reviewer #4: 

1. The title is very critical and important, additionally it covers an extra point of research not covered in previous research.

Response: Thank you for your positive feedback regarding the title. I have carefully reviewed the content and can confirm that all the relevant aspects are indeed covered. If there are any other concerns or suggestions, please do let us know. We are open to feedback and will take your comments seriously.

2. Methodology: Please mention more specifically why select these two hospitals

Response: We deeply appreciate the valuable feedback you've given. The hospitals in Wuhan represent the palliative care triage levels in provincial cities, while those in Huangshi reflect the triage levels in regional cities. The number of beds in these two hospitals ranges from 1,800 to 3,300.

3. Data collection process very short give more clarification about actual process regarding the duration to fill every sheet, additionally how you decrease the bias during data collection process, Moreover, more clarification about the role of co-investigators

Response: Your feedback is greatly valued and has been instrumental to our work. Data collection methods The researcher contacted the hospital, secured approval from its nursing department, and held face-to-face meetings with the potential participants and hospital director to explain the study’s purpose, risks, and benefits. Trained research assistants from each hospital collected data following a standardised protocol and served as co-investigators. Data were gathered via an online survey platform (https://www.wjx.cn/) . To ensure data integrity, each questionnaire could be submitted only once per IP address. Participants were assured of confidentiality, anonymity, and their right to withdraw at any time without consequences.

4.The Palliative Care Graded Diagnosis and Treatment Knowledge, Attitudes, and Practices tool please tension about this paragraph “In the knowledge dimension, participants were primarily surveyed on their knowledge of palliative care graded diagnosis and treatment, rated on a 5-point scale (1 = strongly disagree, 5 = strongly agree). Higher scores indicate more positive attitudes towards palliative care graded diagnosis and treatment. The behavior dimension consists of 13 items, rated from 1(very rarely) to 5(always), with higher scores indicating more proactive engagement in palliative care graded diagnosis and treatment. “ please more clarification about the three dimensions in this tool about how many items in each dimension because you mention knowledge dimension and you mention this dimension used to assess attitude and you didn’t mention the attitude dimension itself and how many items????.

Response: Thank you for your valuable feedback. In the knowledge dimension, contains 11 entries with single-choice answers, participants were primarily surveyed on their knowledge of palliative care triage. The attitude dimension had 22 items, rated on a 5-point scale (1 = strongly disagree, 5 = strongly agree). Its total score ranged from 22 to 110, with higher scores indicate more positive attitudes towards palliative care triage. The behavior dimension consists of 13 items, rated from 1( very rarely) to 5(always),had a total score range of 13 to 65. Higher scores indicating more proactive engagement in palliative care triage.

5、In the General Self-Efficacy Scale (GSES), developed by Ralf Schwarzer, was used to measure nurses’ self-efficacy , you didn’t mention the Ralf Schwarzer article in the reference section

Response: Your constructive feedback is highly appreciated. 34. Zhang JX, Schwarzer R. Measuring optimistic self-beliefs: a Chinese adaptation of the general self-efficacy scale. Psychologia: An International Journal of Psychology in the Orient. 1995;:174–81.

6. What about pilot sample ?

Response: Prior to the formal implementation of this study, we conducted a pilot study to assess the feasibility of the questionnaire, the response of the participants, and the effectiveness of the data collection process. The pilot study was carried out in June 2023, involving 60 nurses from the same two hospitals that were included in the main study. The purpose of the pilot was to identify and address any issues that could potentially affect the outcomes of the formal study, including the difficulty of understanding the questionnaire, the level of participant engagement, and the integrity of the data collected.The results of the pilot study indicated that the questionnaire was generally clear and that participants were able to understand and complete all items. However, we observed that several items in the General Self-Efficacy Scale required further clarification to ensure that all participants could accurately comprehend them. Additionally, the preliminary analysis of the pilot data suggested that the sample size might not be sufficient to detect subtle differences between self-efficacy, compassion fatigue, and job satisfaction. Consequently, through a calculation of the required sample size, we decided to increase the sample size for the formal study to 900 nurses to enhance the statistical power.

7.This manuscript requires additional editing and proofreading to correct detected errors

Response: Thank you for your constructive feedback regarding the manuscript. We have taken your comments seriously and have conducted additional editing and proofreading to address the errors that were detected. We are confident that the manuscript has been significantly improved in terms of language accuracy and overall clarity.We would be grateful if you could review the revised manuscript and provide us with any further suggestions or comments you may have to ensure the highest quality of our submission.

8.Please add DOI in some references

Response: Thank you for your attention to detail. We have added the DOIs to the relevant references as suggested. Additionally, we have conducted a thorough check to ensure that all references now include the appropriate DOI information where available.

---

## [Decision Letter · Decision Letter 2]

4 Jun 2025

Dear Dr. Zhang,

Thank you for submitting your manuscript to PLOS ONE. After careful consideration, we feel that it has merit but does not fully meet PLOS ONE’s publication criteria as it currently stands. Therefore, we invite you to submit a revised version of the manuscript that addresses the points raised during the review process.

We look forward to receiving your revised manuscript.

Kind regards,

Mohammed Elsayed Zaky, Ph.D

Academic Editor

PLOS ONE

**Journal Requirements:**

Reviewers' comments:

Reviewer's Responses to Questions

**Comments to the Author**

Reviewer #2: All comments have been addressed

Reviewer #5: All comments have been addressed

2. Is the manuscript technically sound, and do the data support the conclusions?

Reviewer #2: Yes

Reviewer #5: Partly

3. Has the statistical analysis been performed appropriately and rigorously?

Reviewer #2: Yes

Reviewer #5: Yes

4. Have the authors made all data underlying the findings in their manuscript fully available?

Reviewer #2: Yes

Reviewer #5: Yes

5. Is the manuscript presented in an intelligible fashion and written in standard English?

Reviewer #2: Yes

Reviewer #5: Yes

**Reviewer #2: ** (No Response)

**Reviewer #5: ** The study addresses a highly relevant and underexplored topic, offering valuable insights into the psychosocial dynamics affecting palliative care practices among nurses. The use of a moderated mediation model enhances the conceptual strength of the manuscript. However, several aspects would benefit from improved clarity, methodological detail, and consistency in reporting. The following specific comments are intended to support the refinement of your manuscript.

• The manuscript inconsistently uses the terms “grading treatment,” “graded diagnosis and treatment,” and “triage.” Therefore standardize terminology throughout. If "graded diagnosis and treatment" is a unique model in China, define it clearly early on and use it consistently (e.g., "graded treatment"). If you intend to use "triage" as a broader concept, clarify its equivalence or difference with "graded treatment."

• Condense the aims into a single, cohesive paragraph. For example: "This study aims to (1) examine the relationship between healthcare professionals’ attitudes and behaviors regarding palliative care grading treatment and the mediating role of self-efficacy, and (2) explore whether self-efficacy moderates the effect of compassion fatigue on well-being at work."

• Improve Background Flow by streamline by grouping related ideas (e.g., barriers to palliative care delivery, role of healthcare professionals, system-level challenges) into more cohesive thematic sections. You can use updated studies in the background “Toqan D, Malak MZ, Ayed A, Hamaideh SH, Al-Amer R. Perception of nurses’ knowledge about Palliative Care in West Bank/Palestine: levels and influencing factors. Journal of palliative care. 2023 Jul;38(3):336-44”.

“Awad B, Batran A, Malak MZ, Ayed A, Shehadeh A, Alassoud B, Ejheisheh MA. Knowledge, attitudes, and self-efficacy regarding palliative care among Palestinian nurses in intensive care units. BMC nursing. 2025 Apr 17;24(1):435”.

• Sharpen the Justification for the Study "To our knowledge, this is among the first studies to assess both the mediating and moderating roles of self-efficacy in the context of palliative care treatment grading in China, filling a critical gap in understanding the psychosocial mechanisms behind healthcare professionals' behaviors."

• Change “To investigates” to “To investigate” in Aims. Change “triage attitude” to “attitudes toward graded treatment” (if this is the consistent term).

• Design ???

• Population and procedure: The text mentions distributing 900 surveys and completing 890, but later also says 900 were distributed and 742 were valid—consider clarifying this sequence for better coherence.

• Ethical approval : Clarify confidentiality procedures: You may elaborate slightly on how identification numbers were used (e.g., coded, encrypted).

• Measurement tools: Cronbach’s alpha placement: Mention the reliability coefficient at the end of each scale section for clarity.

**Do you want your identity to be public for this peer review?** For information about this choice, including consent withdrawal, please see our Privacy Policy

Reviewer #2: No

Reviewer #5: No

---

## [Author Response · Author response to Decision Letter 3]

23 Jun 2025

1�The manuscript inconsistently uses the terms “grading treatment,” “graded diagnosis and treatment,” and “triage.” Therefore standardize terminology throughout. If "graded diagnosis and treatment" is a unique model in China, define it clearly early on and use it consistently (e.g., "graded treatment"). If you intend to use "triage" as a broader concept, clarify its equivalence or difference with "graded treatment."

Response: Thank you for highlighting the importance of consistent terminology in our manuscript. We have carefully reviewed and standardized the terms throughout the text. We have exclusively used "graded diagnosis and treatment" as recommended, and have clarified its definition early in the manuscript to reflect its unique application in the Chinese healthcare context in background. In China, a graded diagnosis and treatment system is implemented, whereby medical cases are categorized based on disease severity and treatment complexity.

2�Condense the aims into a single, cohesive paragraph. For example: "This study aims to (1) examine the relationship between healthcare professionals’ attitudes and behaviors regarding palliative care grading treatment and the mediating role of self-efficacy, and (2) explore whether self-efficacy moderates the effect of compassion fatigue on well-being at work."

Response:Thank you for your valuable feedback. This study aims to (1) investigate the relationship between healthcare professionals' attitudes and behaviors toward palliative care grading treatment and the mediating role of self-efficacy, and (2) explore whether self-efficacy moderates the effect of compassion fatigue on work-related well-being.

3�Improve Background Flow by streamline by grouping related ideas (e.g., barriers to palliative care delivery, role of healthcare professionals, system-level challenges) into more cohesive thematic sections. You can use updated studies in the background “Toqan D, Malak MZ, Ayed A, Hamaideh SH, Al-Amer R. Perception of nurses’ knowledge about Palliative Care in West Bank/Palestine: levels and influencing factors. Journal of palliative care. 2023 Jul;38(3):336-44”.

Response: Thank you so much for your insightful feedback. We have revised the background section of our manuscript in accordance with the references you suggested, which we believe has enhanced the clarity and comprehensiveness of our work. These findings suggest that various socio-demographic factors may significantly impact the level of knowledge regarding palliative care among nurses. As demonstrated in recent research[15].

4�“Awad B, Batran A, Malak MZ, Ayed A, Shehadeh A, Alassoud B, Ejheisheh MA. Knowledge, attitudes, and self-efficacy regarding palliative care among Palestinian nurses in intensive care units. BMC nursing. 2025 Apr 17;24(1):435”.

Response: We appreciate your constructive feedback. Based on the references you provided, we have thoroughly revised the background section of our manuscript. We are confident that these revisions have significantly improved the clarity and comprehensiveness of our work. The research investigated the knowledge, attitudes, and self-efficacy related to palliative care among Palestinian nurses working in intensive care units (ICUs). The results indicated that ICU nurses in the region demonstrated insufficient knowledge of palliative care and exhibited low self-efficacy in its administration. These findings imply that gaps in knowledge and unfavorable attitudes among healthcare professionals may constitute obstacles to the effective implementation of palliative care services[14].

5�Sharpen the Justification for the Study "To our knowledge, this is among the first studies to assess both the mediating and moderating roles of self-efficacy in the context of palliative care treatment grading in China, filling a critical gap in understanding the psychosocial mechanisms behind healthcare professionals' behaviors."

Response: Thank you for your valuable comments. We're sorry for the inaccurate description. We did not fill the relevant research gap. Instead, we only put forward local findings and our own views on this field. After a literature review, we found few articles and views on this topic. We stress this study's innovative assessment of self - efficacy's mediating and moderating roles in palliative care grading. Based on a thorough literature search and analysis, no prior studies have directly focused on this specific aspect. We cited study [53], conducted in 10 Zhejiang compulsory isolation drug rehabilitation centers with 1,197 SUDs patients, where attitudinal behavior affected behavioral intention (β = 0.02) without self - efficacy as a mediator. We have removed the absolute statements that might cause misunderstandings and changed them into more rigorous and objective descriptions. However, research on the impact of self - efficacy on the graded diagnosis and treatment behavior of healthcare professionals is currently limited in quantity. Thus, this study offers significant insights and contributes a new perspective to this area of knowledge.

6�Change “To investigates” to “To investigate” in Aims. Change “triage attitude” to “attitudes toward graded treatment” (if this is the consistent term).

Response: Thank you for your comments on the Aims section and the terminology used in our manuscript. The phrase “To investigates” has been corrected to “To investigate” in the Aims section.The term “triage attitude” has been changed to “attitudes toward graded diagnosis and treatment” to ensure consistency with the terminology used throughout the manuscript.

7�Design ???

Response: Thank you for your valuable feedback. A cross-sectional study was conducted on July 26, 2023, and completed on August 30, 2023.

8�Population and procedure: The text mentions distributing 900 surveys and completing 890, but later also says 900 were distributed and 742 were valid—consider clarifying this sequence for better coherence.

Response: Thank you so much for your insightful feedback. A total of 900 survey forms were distributed. After data cleaning, 742 responses (validity rate of 77.6%) were deemed suitable for data analysis.

9�Ethical approval : Clarify confidentiality procedures: You may elaborate slightly on how identification numbers were used (e.g., coded, encrypted).

Response: Thank you for your valuable feedback. To safeguard individual identities, only numerical identifiers were used on questionnaires, eschewing any identifiable patient data. All participants were assigned unique identification numbers upon enrollment. These numbers were allocated in the sequence of questionnaire completion and stored separately from the informed consent documents.

10�Measurement tools: Cronbach’s alpha placement: Mention the reliability coefficient at the end of each scale section for clarity.

Response: Thank you for your insightful feedback regarding the placement of Cronbach’s alpha in the measurement tools section. We have carefully reviewed our manuscript and confirmed that the Cronbach’s alpha for each tool was indeed placed at the end of the respective tool section. We are grateful for your suggestion, which aligns with our approach and helps to ensure clarity and coherence in presenting the reliability coefficients.

---

## [Decision Letter · Decision Letter 3]

22 Sep 2025

Healthcare Providers' Palliative Care Graded Diagnosis and Treatment Behavior, Attitudes, Self-efficacy, Compassion Fatigue, and Workplace Well-being�A Mediating Moderation Model

PLOS ONE

Dear Dr. Zhang,

Thank you for submitting your manuscript to PLOS ONE. After careful consideration, we feel that it has merit but does not fully meet PLOS ONE’s publication criteria as it currently stands. Therefore, we invite you to submit a revised version of the manuscript that addresses the points raised during the review process.

https://journals.plos.org/plosone/s/submission-guidelines#loc-laboratory-protocols . Additionally, PLOS ONE offers an option for publishing peer-reviewed Lab Protocol articles, which describe protocols hosted on protocols.io. Read more information on sharing protocols at https://plos.org/protocols?utm_medium=editorial-email&utm_source=authorletters&utm_campaign=protocols .

We look forward to receiving your revised manuscript.

Kind regards,

Majed Sulaiman Alamri, PhD

Academic Editor

PLOS ONE

Journal Requirements:

Reviewers' comments:

Reviewer's Responses to Questions

**Comments to the Author**

Reviewer #5: All comments have been addressed

Reviewer #6: All comments have been addressed

Reviewer #7: (No Response)

2. Is the manuscript technically sound, and do the data support the conclusions?

Reviewer #5: Yes

Reviewer #6: Yes

Reviewer #7: Yes

3. Has the statistical analysis been performed appropriately and rigorously?

Reviewer #5: Yes

Reviewer #6: Yes

Reviewer #7: Yes

4. Have the authors made all data underlying the findings in their manuscript fully available?

Reviewer #5: Yes

Reviewer #6: Yes

Reviewer #7: Yes

5. Is the manuscript presented in an intelligible fashion and written in standard English?

Reviewer #5: Yes

Reviewer #6: Yes

Reviewer #7: Yes

Reviewer #5: the authors adequately addressed my comments raised in a previous round of review and the language is clear, correct, and unambiguous

Reviewer #6: The revised manuscript (PONE-D-24-35851R3) has successfully addressed all comments. The changes are substantial and significantly improve the quality, clarity, and academic rigor of the paper. The most important change, the consistent use of "graded diagnosis and treatment", has been implemented correctly.

The manuscript is now in a much stronger position for acceptance. The remaining items are very minor typographical edits that can be caught in a final proofread before resubmission.

Reviewer #7: Dear authors,

The topic you have done is important

Some comments

First, tracking and having two versions within one document was time-consuming; you should provide only what you amended, and those with accepting changes in font and colour should not appear.

Some grammar needs to be checked; for example, self-efficacy is an individual's ability of to execute and control the influences of their environment at a predetermined level [19].

900 healthcare providers or nurses? The abstract mentioned healthcare providers. Also, in discussion healthcare providers.

Clearly distinguish which subscales were analysed; only attitude was seen in the analysis. (KAP?)

In discussion, self-efficacy’s protective role is repeated across multiple paragraphs. Overly repetitive

Maybe you need a paragraph connecting findings to training and policy interventions more explicitly.

The last version is improved compared to the original one.

**Do you want your identity to be public for this peer review?** For information about this choice, including consent withdrawal, please see our Privacy Policy

Reviewer #5: No

Reviewer #6: No

Reviewer #7: No

---

## [Author Response · Author response to Decision Letter 4]

30 Oct 2025

Some grammar needs to be checked; for example, self-efficacy is an individual's ability of to execute and control the influences of their environment at a predetermined level [19].

Thank you very much for your careful reading and for pointing out this grammatical error. We have corrected the sentence.

900 healthcare providers or nurses? The abstract mentioned healthcare providers. Also, in discussion healthcare providers.

Thank you for highlighting this inconsistency. We have now harmonised the description of the study population throughout the manuscript.

Clearly distinguish which subscales were analysed; only attitude was seen in the analysis. (KAP?)

Thank you for your helpful feedback. Only the attitude subscale was retained as a predictor: the knowledge dimension showed a severe ceiling effect and near-zero variance, rendering it unsuitable for regression, whereas the behaviour dimension served as the study’s outcome variable and therefore could not be included as an independent or mediating factor. All inferential tests were thus performed on attitude scores.

In discussion, self-efficacy’s protective role is repeated across multiple paragraphs. Overly repetitive.

Thank you for highlighting this issue. The apparently repetitive references to the “protective role” of self-efficacy actually appear in four distinct analytical contexts:

p. 20, para. 2: attitude–self-efficacy link (H1)

p. 21, para. 1: behavior–self-efficacy link (H2)

p. 22, para. 1: attitude → self-efficacy → behavior mediation (H3)

p. 23, para. 1: self-efficacy moderating compassion-fatigue → well-being (H4)

To eliminate overlap we have now retained the term “protective” only once, when first introducing H2.

Maybe you need a paragraph connecting findings to training and policy interventions more explicitly.

Thank you very much for your invaluable feedback.We have restructured the paragraph accordingly.

Self-efficacy emerged as a key determinant and mediator of providers’ graded-care participation, guiding future training design.Translating psychosocial evidence into scalable action, through targeted training and interventions, boosting their confidence and decision-making abilities can effectively encourage their active participation in complex situations and improve the quality of graded diagnosis and treatment.

---

## [Decision Letter · Decision Letter 4]

3 Nov 2025

Healthcare Providers' Palliative Care Graded Diagnosis and Treatment Behavior, Attitudes, Self-efficacy, Compassion Fatigue, and Workplace Well-being�A Mediating Moderation Model

PONE-D-24-35851R4

Dear Dr. Zhang, 

We’re pleased to inform you that your manuscript has been judged scientifically suitable for publication and will be formally accepted for publication once it meets all outstanding technical requirements.

Kind regards,

Majed Sulaiman Alamri, PhD

Academic Editor

PLOS ONE

Additional Editor Comments (optional):

Reviewers' comments:

Reviewer's Responses to Questions

**Comments to the Author**

Reviewer #5: All comments have been addressed

Reviewer #6: All comments have been addressed

2. Is the manuscript technically sound, and do the data support the conclusions?

Reviewer #5: Yes

Reviewer #6: Yes

3. Has the statistical analysis been performed appropriately and rigorously?

Reviewer #5: Yes

Reviewer #6: Yes

4. Have the authors made all data underlying the findings in their manuscript fully available?

Reviewer #5: Yes

Reviewer #6: Yes

5. Is the manuscript presented in an intelligible fashion and written in standard English?

Reviewer #5: Yes

Reviewer #6: Yes

Reviewer #5: the authors do all comments. The manuscript described a technically sound piece of scientific research with data supported the conclusions. the language is clear, correct, and unambiguous. no comments

Reviewer #6: The study is well-conceived and makes a valuable contribution to the literature on palliative care. The cross-sectional design and self-report measures are appropriately acknowledged as limitations. The manuscript is methodologically sound, ethically compliant, and presents its findings clearly.

I have no further concerns and recommend acceptance for publication.

**Do you want your identity to be public for this peer review?** For information about this choice, including consent withdrawal, please see our Privacy Policy

Reviewer #5: No

Reviewer #6: No

---

## [Editor Report · Acceptance letter]

PONE-D-24-35851R4

PLOS ONE

Dear Dr. Zhang,

I'm pleased to inform you that your manuscript has been deemed suitable for publication in PLOS ONE. Congratulations! Your manuscript is now being handed over to our production team.

Kind regards,

on behalf of

Prof. Majed Sulaiman Alamri

Academic Editor

PLOS ONE